# Expected Reward Prediction, with Applications to Model Routing

## Abstract

Reward models are a standard tool to score responses from LLMs. Reward models are built to rank responses to a fixed prompt sampled from a single model, for example to choose the best of $n$ sampled responses. In this paper, we study whether scores from response-level reward models lifted to score a *model's* suitability for a prompt, prior to seeing responses from that model. Specifically, we show that it is straightforward to predict the expected reward that an LLM would earn from the reward model under repeated sampling. Further, we show that these expected reward predictions are precise and discriminative enough to support an application to a model routing protocol that routes prompts to models at inference time to maximize reward while controlling computational cost. We demonstrate the performance of this routing procedure on the open-perfectblend dataset, using a model pool composed of Llama3.1-Instruct 8B/70B, Gemma2-IT 9B/27B, and Gemma1-IT 7B models. Our simple expected reward prediction–based routing (ERP) outperforms baselines that route prompts to models with the best average performance within each prompt's category, and explains the success of more complex routing protocols that implicitly estimate an expected reward. Our approach has the added advantage of being trivially extensible as new models are added to the pool.

## 1 Introduction

Reward models are commonly used in large language model (LLM) alignment, sampling, and evaluation (see, e.g. Christiano et al., 2017; Stiennon et al., 2020; Gao et al., 2023; Wang et al., 2024a). A reward model is a function that takes a prompt $x$ and a response $y$ and returns a score $r(x, y)$ quantifying how good the response is for the prompt. Notice that this makes no reference to language models—the reward is a property of text, not of a generative model. In this paper, we are interested in understanding how good a *model* is for a given prompt. More precisely, we are interested in lifting a reward function on *responses* to a reward function on *models*.

The aim here can be understood as producing a LLM-level reward function that predicts *a priori* how well a random sample from the LLM can be expected to perform in response to the prompt $x$. Such a priori predictions would be useful for a number of inference time operations, such as model routing and prompt modification. Moreover, obtaining model-level rewards from response-level rewards would be especially convenient since there has already been enormous research and development effort put into creating high quality response-level rewards (Lambert et al., 2024). In this paper, we formalize the a priori reward of a language model $\pi$ as $\mathbb{E}_{Y \sim \pi(y|x)}[r(x, Y)]$, the expected value of the response level reward of a random sample drawn from the LLM. The key question is whether it is possible to predict this quantity.

It is not clear that predicting the expected reward *a priori* is actually possible, for at least two reasons. First, there is no guarantee that the generating LLM is well-behaved enough for its prompt-specific behavior to be easily predicted. There are two potentially unpredictable elements: (1) the sensitivity of the model to specifics of the prompt, and (2) the distribution of responses that a non-deterministic model can give to a prompt. If either aspect of the generating model is not well-behaved, it may not be practical to predict $\mathbb{E}_{Y \sim P(y|x)}Y[r(x, Y)]$, which summarizes a large range of behavior of the model. Secondly, the large majority of reward models used in practice are trained to encode the *relative* quality of two responses to the same prompt. That is, theoretically reward models only

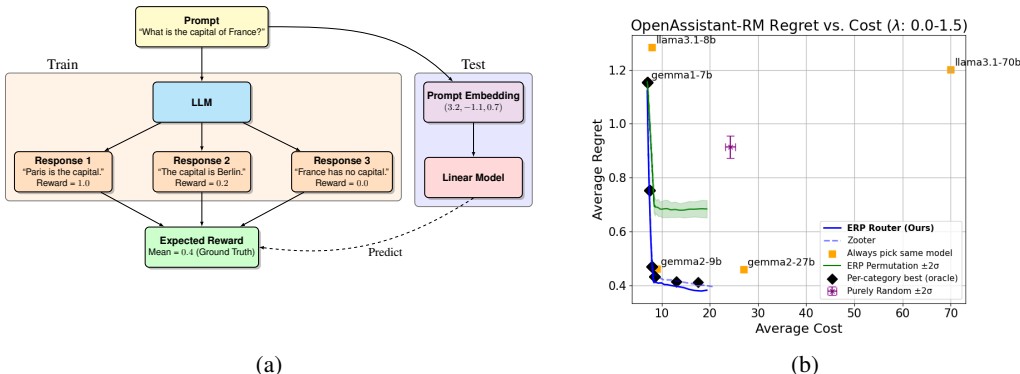

(a)  (b)

Figure 1: (Left) **Expected reward prediction workflow.** We train a linear model on an embedding of the prompt to predict the expected reward that the model would earn responding to the prompt. (Right) **Expected reward prediction (ERP) can be used to route queries to the best, cost-effective model for that query.** The Pareto frontier (blue) shows that an ERP-based routing policy dominates baselines that don't discriminate between models in the context of each prompt, regardless of cost sensitivity. Additional regret-cost tradeoff plots for other RMs can be found in Figure 6.

guarantee that $r(x, y_1) - r(x, y_0)$ is a meaningful quantity for all pairs of responses $y_0, y_1$, but allow an arbitrary prompt-dependent value to be added to each reward (which cancels out when we take the difference). A model's expected reward for a given prompt depends on this arbitrary value, and thus need not be learnable even in principle.[1] Thus, to the extent that prompt-wise expected reward is predictable, we would expect it to be a consequence of the implementation of the reward model as a fine-tuned pretrained language model.

For these reasons, the scope of our contribution is primarily empirical. The main finding of this paper is that it is in fact possible to predict the expected reward with high fidelity using realistic language models and reward functions. Moreover, we find that this prediction is possible even using an astonishingly simple model: a linear probe trained on an off-the-shelf embedding representation of the prompt (Figure 1(a)). Finally, we demonstrate that this predictive power is not merely an artifact of identifying universally "good" or "bad" prompts: we show that predicted expected reward can be used to discriminate between models that would provide better or worse responses to the specific prompt. Specifically, we demonstrate this with an application to model routing. In the model routing application, we take in a prompt $x$ and decide which model it should be optimally served to. We find that using lifted rewards leads to large improvements over any fixed model, particularly in applications where querying each model has variable cost (Figure 1(b)).

## 2 PRELIMINARIES

### 2.1 LANGUAGE MODELS

In this paper, we view a large language model $\pi$ as a kernel that maps prompts $x$ to probability distributions $\pi(Y \mid x)$ over responses $y$. Our interest is in predicting aspects of the response *a priori*. Formally, this is equivalent to characterizing (properties of) the probability distribution $\pi(Y \mid x)$ from the prompt. In particular, we will be interested in learning functions of the form $\mathrm{ER}_\pi(x) := \mathbb{E}_{Y \sim \pi(Y|x)}[r(x, Y)].$

### 2.2 REWARD MODELING

A central problem in generative language model evaluation is scoring open-ended outputs that vary on a large number of dimensions that are meaningful to humans. Pre-specifying these dimensions and scoring responses along each would be onerous, especially for hard-to-define concepts like "helpfulness". Reward models avoid this problem by inferring an overall notion of "goodness" that is

---

[1]Formally, the expected value is not identified under the Bradley-Terry model.

revealed by preference data. Preference data has the form $(x, y_-, y_+)$, where $x$ is the input prompt, $y_+$ and $y_-$ are the preferred and dispreferred candidate responses. A reward model translates this preference-level data into a response-wise score $r(x, y)$, so that for a given prompt, $r(x, y) > r(x, y')$ implies that $y$ is more likely to be preferred to $y'$ by raters that generated the preference data.

The standard approach to reward modeling is to do this translation by maximizing the Bradley-Terry log-likelihood.

$$r^*(x, y) = \arg\max_r \mathbb{E}_{x \sim p_{\mathcal{X}}(x)}[\log \sigma(r(x, y_+) - r(x, y_-))] \tag{1}$$

A useful reward score $r(x, y)$ is a complex function of $x$ and $y$, requiring semantic understanding of the prompt and the appropriateness of the response. For this reason, reward models are usually obtained by fine-tuning a large language model, which bake in the requisite semantic understanding (see, e.g., Wang et al., 2024a, for a review).

Initially, reward models were trained on narrow preference datasets in the context of specific tasks, to target relatively narrow notions of "goodness", such as harmfulness and helpfulness in assistant conversations (Bai et al., 2022), or the usefulness of a summary in summarization tasks (Stiennon et al., 2020). However, increasingly, general purpose reward models are being trained that can measure "goodness" on a variety of downstream tasks. Reward models of this type are compared on the RewardBench benchmark Lambert et al. (2024), and high-performing general purpose reward models appear on the RewardBench leaderboard (`https://huggingface.co/spaces/allenai/reward-bench`).

### 2.3 Experimental Setup

The aim of this paper is to understand whether, and how, the expected reward of language model $\pi$ for reward function $r$ can be predicted from a prompt.

**Prompt Dataset.** To explore the predictability of expected rewards, we use a diverse dataset of prompts for which we expect there to be variability in the quality of model responses, both within models (i.e., for a given model, the model will give better responses to some prompts than other prompts, in expectation) and between models (i.e., for a given prompt, some models will give better responses than other models, in expectation).

Specifically, we use the open-perfectblend dataset (Labonne, 2024), which is an open source near-replication of the mixture of datasets introduced in Xu et al. (2024). The dataset was initially designed as a diverse supervised fine-tuning training set, with prompts from datasets focused on general chat capabilities, instruction following, math reasoning, and code reasoning (see documentation for Labonne (2024) for full details).

For our experiments, we sample 1000 prompts at random from each of the 4 mentioned categories in open-perfectblend for a total of 4000 prompts. The general chat data in open-perfectblend is formatted as multi-turn conversations, but here we focus on reward for single-turn responses, so we truncate to the first user query in the chat conversation.

**Large Language Models.** We study expected reward predictability across several LLM sizes in open weight model families. We focus on the following instruction-tuned generating models: **Llama 3.1-IT (8B, 70B)** (Dubey et al., 2024), **Gemma 2-IT (9B, 27B)** (Team et al., 2024b), **Gemma 1-IT (7B)** (Team et al., 2024a). When sampling from each model, we use standard autoregressive temperature sampling, where we sample one token at a time, conditional on all that came before. For all models, we set the temperature to 1.0.

**Reward Models.** We study the predictability of the expected scores from three reward models: **OpenAssistant-RM**[2], **GRM-2B-RM**[3] (Yang et al., 2024), **InternLM-RM**[4] (Cai et al., 2024). These reward models were all trained using the Bradley-Terry objective to encode complex, aggregate

---

[2] `https://huggingface.co/OpenAssistant/reward-model-deberta-v3-large-v2`
[3] `https://huggingface.co/Ray2333/GRM-Gemma-2B-rewardmodel-ft`
[4] `https://huggingface.co/internlm/internlm2-7b-reward`

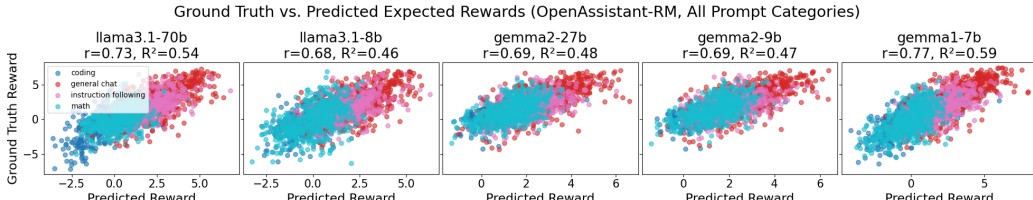

Figure 2: Expected reward is predictable both within and between prompt classes, across model families and sizes. Each point shows the predicted and empirical expected reward for a prompt from the test split of the open-perfectblend dataset. Points are colored by their category. Predictive power is measured by $R^2$; roughly, the variance explained by the predictions. Additional reward plots for other RMs can be found in Figure 7.

human preferences, performed well on RewardBench for their respective sizes, and were chosen to be small due to computational restrictions.

## 2.4 RELATED WORK

**Distributional Property Prediction.**   The key idea in this paper is to predict a priori a distributional property of an LLM output (namely, the expected reward) from the prompt alone. The idea of predicting distributional aspects has also occurred in some other contexts. For example, Kossen et al. (2024) train linear probes to predict the entropy of the output distribution in the context of hallucination detection. Wang et al. (2024c) also train a predictor for the expected reward as a function of the prompt as a component of a modified RLHF algorithm. In contrast to the present paper, they do not attempt to quantify the quality of this predictive model.

**Model Routing Methods.**   Model routing has been an active area of research for cost-effective use of LLMs. Approaches include preference model–based routing (Ong et al., 2024), to which we made a comparison above; and cascade-based routing, were models are sequentially queried until an acceptable answer is found, as judged by a scoring model (Chen et al., 2023) or a self-verification mechanism (Madaan et al., 2023). Other approaches close to ours also use predictions of model-wise response quality (Nguyen et al., 2024; Liu et al., 2024), but focus on cases where responses can be judged in terms of binary accuracy. The most similar work to ours that indirectly performs reward prediction is the Zooter method proposed by Lu et al. (2023), which trains a router to predict the winning model's distribution (i.e., response with highest reward). The output logits of this network can be interpreted as a form of reward prediction. Our work shows that the direct prediction of expected reward at the model level is equally effective, more scalable, and a likely explanation for the effectiveness of reward prediction-based routers such as Zooter.

## 3 PREDICTABILITY OF THE EXPECTED REWARD

To begin, we report on our experiments studying the predictability of expected reward across generating models and reward models. The key finding is that, in practice, expected rewards can be predicted with remarkable precision using a linear model built on top of an external embedding of the prompt. That is, the a priori prediction problem is indeed solvable, and, moreover, the prediction can be done with a simple linear model.

**Task Setup.**   Given a prompt $x$, a language model $\pi(Y \mid x)$, and a reward model $r$, our goal is to predict the expected reward that would be assigned to responses sampled from the model,

$$\mathrm{ER}_\pi(x) := \mathbb{E}_{Y \sim \pi(Y|x)}[r(x, Y)]. \tag{2}$$

Here, our goal is to train a model that directly predicts $\mathrm{ER}_\pi(x)$ from $x$, without the need to take repeated samples at inference time.

Table 1: $R^2$ of expected reward prediction within each open-perfectblend category of prompts using OpenAssistant-RM. For most categories, and for most models, the predictor explains variation in rewards, even after conditioning on the question category, suggesting that expected rewards give a finer-grained view of model capabilities. Additional tables with $R^2$ values for other RMs can be found in Table 2.

| Model Name | Aggregate | Coding | Math | Instruction Following | General Chat |
|---|---|---|---|---|---|
| **llama3.1-70b** | 0.54 | 0.41 | 0.19 | 0.25 | 0.42 |
| **llama3.1-8b** | 0.46 | 0.26 | 0.24 | 0.23 | 0.39 |
| **gemma2-27b** | 0.48 | 0.37 | 0.30 | 0.27 | 0.45 |
| **gemma2-9b** | 0.47 | 0.34 | 0.31 | 0.25 | 0.46 |
| **gemma1-7b** | 0.59 | 0.50 | 0.34 | 0.25 | 0.49 |

To set up the learning problem, we build datasets $\mathcal{D}_\pi = \{(x_i, \widehat{\mathrm{ER}}_\pi(x_i))\}_{i=1}^N$ by brute force. For each prompt $x \in \mathcal{X}$, we sample $K$ responses $\{y_i^{(k)}\}_{k=1}^K$ from the language model, compute the reward $r(x_i, y_i^{(k)})$ for each response, and then record the empirical mean reward $\widehat{\mathrm{ER}}_\pi(x_i) = \frac{1}{K}\sum_{k=1}^K r(x_i, y_i^{(k)})$ as the target. For all experiments here, we use $K = 32$. The learning task is to predict label $\widehat{\mathrm{ER}}_\pi(x_i)$ from the prompt $x_i$.

We split these datasets into training and test sets in a $50/50$ manner stratified across prompt categories. Namely, in each category, we delegate half the prompts to the training set and the remaining half to the test set.

**Linear Models.** We train models to predict the empirical mean reward $\widehat{\mathrm{ER}}_\pi(x)$ from each prompt $x$ with linear models that take the a fixed-length vector representation, or embedding $v(x)$ of the prompt $x$ as input. We use ridge-regularized linear models that solve

$$\arg\min_\theta \mathbb{E}_{X \sim p_{\mathcal{X}}(X)}[(\theta^\top v(X) - \widehat{\mathrm{ER}}_\pi(X))^2] + \beta\|\theta\|_2^2 \tag{3}$$

For the embedding representation $v(x)$, we use **gte-large-en-v1.5**, an off-the-shelf pre-trained embedding model with embedding dimension 1024 (Zhang et al., 2024). We chose this embedding model due to being lightweight at under $0.5$ billion parameters and its strong performance (rank 33) on the MTEB leaderboard (Muennighoff et al., 2022). We set $\beta = 1$ to optimize $R^2$ performance and prevent overfitting.

**Results.** In Figure 2, we summarize the predictive performance of linear reward prediction models on open-perfectblend, both in aggregate and within prompt categories[5]. The results show that expected rewards from our chosen reward models are indeed predictable, and linear predictions from embeddings can capture a substantial fraction of the variation in held-out test data, both within and between prompt categories in open-perfectblend.

Notably, for most categories, the predictions and targets have relatively well-behaved distributions, and predictive performance is not easily explained away, for example, by obvious artifacts in the data like a large fraction of refusals to answer. The predictability here suggests that even within specific categories, variation in each model's capabilities (at least as measured by the reward models) across fine-grained prompt classes is easily linearly represented in terms of the general-purpose prompt embedding.

There are some exceptions to this pattern in the predictions of GRM-2B-RM rewards. Specifically $R^2$ values within the math category are low for all models except Llama-3.1 70B, and are similarly low in the instruction following category for the Llama-3.1 models. Nonetheless, aggregate reward remains highly predictable, because the GRM-2B-RM groups scores for these categories relatively tightly in the full range of scores.

**Discussion.** As we have emphasized, the predictability of expected reward at all is somewhat surprising. One possible explanation is that modern reward models are finetuned from pre-trained

---

[5]We do not use category information in any way at training time, we only split the test set by category for displaying results.

language models on datasets where there are typically many prompts and only a relatively small number of responses to each prompt, which may bias the model to using a single reward scale to facilitate generalizing across prompts. Another possibility is that some responses, such as refusals to answer, are assigned low rewards in training data regardless of prompt, creating common structure between prompts that the reward models can exploit. However, this is speculation, and the phenomenon deserves more precise investigation. It would be useful to better understand the classes of reward models for which we can expect expected reward prediction to be effective, and to perhaps consider training strategies that actively encourage this property in reward scores.

# 4 APPLICATIONS TO MODEL ROUTING

We have now established that prompt-level expected rewards from our two reward models have systematic predictable structure within each model. Here, we show that these predictions are sufficiently precise to support practical comparisons between models. To do this, we use the predictable structure in prompt-level expectations to design a simple but effective proof-of-concept model routing algorithm. The success of this algorithm is evidence that predicted expected rewards go beyond classifying generally "good" or "bad" prompts, but actually lift response-level scores to model-level scores that can be used to discriminate between them.

## 4.1 MODEL ROUTING SETUP

**Task Definition.** Suppose we have a pool of $M$ models, $\Pi := \{\pi_i\}_{i=1}^{M}$. When we receive a prompt query $x$, we can choose a model to which to route the prompt, and return the response from that model. The goal is to choose a model that can generate a high-quality response cheaply.

Formally, for a given prompt $x$, model $i$ can generate a response $Y_i$ for a computational cost $c(i, x, Y_i)$, earning a reward $r(x, Y_i)$. We aim to learn a routing protocol that maps prompts to models to maximize the expected reward while also maintaining a low computational cost. That is, we want a router $\rho : \mathcal{X} \mapsto \{0, 1, \cdots, M-1\}$ satisfying the constrained optimization problem

$$\underset{\rho}{\arg\max} \quad \mathbb{E}_{x \sim p_{\mathcal{X}}(x)} \mathbb{E}_{Y \sim \pi_{\rho(x)}(Y|x)}[r(x, Y)],$$

$$\text{subject to} \quad \mathbb{E}_{x \sim p_{\mathcal{X}}(x)} \mathbb{E}_{y \sim \pi_{\rho(x)}(Y|x)}[c(\rho(x), x, Y)] \leq C,$$

(4)

where $p_{\mathcal{X}}$ is a uniform distribution over a space of prompts $\mathcal{X}$, and $C$ is the computational budget.

For simplicity, we focus on the case where the cost function $c(i, x, y)$ only depends on the model $i$ and is proportional to the number of parameters in model. This is a reasonable first order approximation, though there are of course substantial subtleties related to expected generation length, details of model architectures, the relative expense of FLOPS vs. memory, and so forth.

**Routing Evaluations.** We evaluate router quality by the regret it incurs relative to an optimal (oracle) policy for assigning prompts to models. For a given prompt $x$, the regret of (deterministic) policy $\rho$ is

$$\mathcal{R}(m, x) := \max_{i \in \Pi} \mathbb{E}_{Y \sim \pi_i(Y|x)}[r(x, Y)] - \mathbb{E}_{Y \sim \pi_{\rho(x)}(Y|x)}[r(x, Y)].$$ (5)

**Preference-Based Model Routing.** Model routing has been approached before in Ong et al. (2024) as a preference learning problem, where the data have the form $(x, p_+, p_-)$, where $x$ is the prompt, $p_+$ is the winning model with the higher reward, and $p_-$ is the losing model with the lower reward. It is assumed that this pairwise comparison data exists for all pairs of models in $\mathcal{P}$. A preference model is then trained on the preference data, using an objective similar to equation 1, which maps prompts to a score for each model that can be used to rank the models and route the prompt at inference time.

**Expected Reward Prediction (ERP)-Based Routing.** Here, we propose a straightforward alternative that leverages the predictability of expected rewards. For each model in $\Pi$, we train a linear predictor using eq. (3) to predict expected rewards. In the simplest policy, we then just route each prompt $x$ to the model with the highest predicted expected reward for that prompt.

However, routing to the highest reward prediction doesn't incorporate the budget constraint in eq. (4). To account for cost, we introduce a parameter $\lambda$ controlling the relative importance of cost and

response-level reward and define the cost-adjusted policy as

$$\rho_\lambda = \max_{\rho'} \mathbb{E}_{x \sim p_{\mathcal{X}}(x)} \mathbb{E}_{Y \sim \pi_{\rho'(x)}(Y|x)} [r(x, Y) - \lambda(c(\rho'(x), x, Y) - C)]. \tag{6}$$

This expected reward prediction routing method has some intrinsic advantages. Of particular note is that the expected reward prediction models can be trained on each model separately, rather than requiring pairwise comparisons between samples from different models. This means that if new models are added to the model pool, one only needs to train a single prediction model for each new model, rather than training a new preference model that requires pairwise comparisons between each model in the old and new sets. In general, data requirements for this method only scale as the number of models, whereas data requirements for direct preference modeling methods scale as the square of the number of models.

This simplicity is not free. It essentially requires that the expected reward suffices to characterize the overall performance of the model. This might not be true, for example, if one of the LLMs produced low quality samples with high probability, but extremely high quality samples with low probability—in this case, the expected reward would be high even though the typical response is poor. However, if LLMs tend to produce samples with somewhat similar rewards, then the expected reward is a reasonable way of summarizing the overall distribution of outputs. The following result gives a particular formalization of this idea:

**Proposition 4.1.** *Let $\pi_0(r(x, Y) \mid x)$ and $\pi_1(r(x, Y) \mid x)$ be the distributions of rewards under models $\pi_0, \pi_1$ respectively. Suppose these distributions are both $\sigma^2$-subgaussian. Then, if $Y_0 \sim \pi_0$ and $Y_1 \sim \pi_1$, we have*

$$\Pr(r(x, Y_1) > r(x, Y_0)) \geq 1 - e^{-\frac{(ER_{\pi_1}(x) - ER_{\pi_0}(x))^2}{4\sigma^2}}$$

*Proof.* Observe that $r(x, Y_1) - r(x, Y_0)$ is $2\sigma$-subgaussian. The result follows immediately. $\square$

In words, this says that the difference in expected rewards relates directly to the win rate of model $\pi_1$ vs $\pi_0$ on the prompt $x$. Thus, in the particular case where the reward distributions are relatively concentrated, routing the model with highest expected reward is a good approximation for routing to the model with the highest overall win rate.

### 4.2 WARMUP: PAIRWISE ERP VS PREFERENCE-BASED WIN PREDICTION

Consider the binary classification task of predicting, for each prompt, which of the two models will generate a better response, according to each of our reward models. We take the binary label to be $\mathbb{1}[r(x, Y_1) > r(x, Y_0)]$, where we produce these labels by using generations from each model. Figure 3 shows the AUROC for the ERP-based binary predictor computed by thresholding $\sigma \left( \widehat{ER}_{\pi_1}(x) - \widehat{ER}_{\pi_0}(x) \right)$ across both reward models on the open-perfectblend test data. We show results for two reward models and each pair of LLMs. The main observation is that the AUROCs are high, meaning we can predict prompt-wise model preferences from the predicted expected rewards.

As a point of comparison, Figure 3 also shows the AUROC for binary predictors trained directly on pairwise reward comparison data for each *pair* of models, using logistic regression on the same input representation $v(x)$. Notice that the values are no better than the expected reward prediction routing. That is, ERP routing, based only on single-model rewards, matches the pairwise classifier's performance.

### 4.3 ROUTING EXPERIMENT

We now present our main experimental results for the model routing application. In this experiment, we route each prompt in our open-perfectblend test split according to a routing policy based on expected reward prediction (ERP) as well as several baseline policies. Across a range of values for the reward-cost exchange-rate parameter $\lambda$, we show that the ERP-based policy improves on the baselines in terms of regret and cost. Furthermore, in a separate set of experiments, we demonstrate that ERP also provides an improved regret-cost tradeoff in the setting of *verifiable rewards*. Details on the verifiable reward experiments are provided in Appendix A.1.

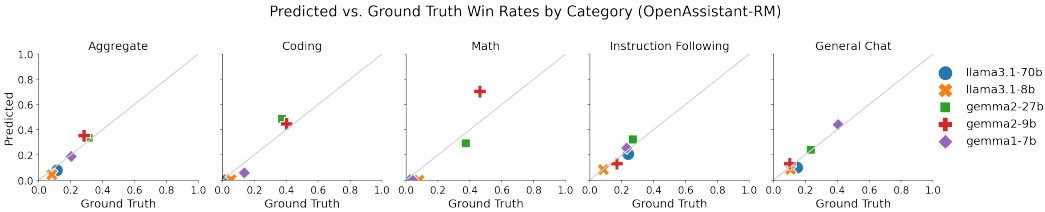

Figure 3: Comparison of our expected reward predictor (ERP) to logistic regression when predicting pairwise model wins using OpenAssistant-RM. AUCs of each win prediction method are nearly identical, but ERP only requires one predictor per model, instead of a separate logistic regression model for each model pair. Additional AUROC figures for other RMs are available in Figure 8.

Figure 4: ERP comparisons accurately predict win rates within each category. Notably, no single model dominates across all categories, and ERP captures model-wise differences. Additional win rate figures for ERP using other RMs can be found in Figure 9.

**Variability in model performance.** We first establish that there is enough variability in model performance across categories to warrant prompt-wise routing. Figure 4 shows that within most open-perfectblend categories, for each reward model, there is diversity across the models that provide the best response to each prompt among the five models in our pool. Notably, GRM-2B-RM's scores are rather lopsided in some categories, a fact that we will return to in our results. However, broadly, this winner diversity establishes that, even within prompt categories, there is an opportunity to improve served responses by routing each prompt to a predicted best model. In addition, as in the pairwise comparison above, predicted win rates, according to ERP, closely mirror ground truth.

**Routing policy and baselines.** Our test policies are:

- **ERP-based.** Directly predict the cost-adjusted reward for each model, and choose the model that maximizes it. When the cost function only depends on the model—as in our experiments—the policy is

$$\rho_\lambda(x) = \arg\max_{i \in \Pi} \widehat{\text{ER}}_{\pi_i}(x) - \lambda c(i). \tag{7}$$

- **Zooter.** Collectively predicts the softmax distribution for the best response. That is, given a prompt $x$ and reward score for each model's response $\{r_i\}_{i=1}^M$, Zooter learns a network $\mathcal{Z}$ that maps $x$ to an $M$-dimensional probability distribution $\{p_i\}_{i=1}^M$ trained to minimize the KL-divergence with respect to the ground-truth winning response distribution softmax$(r_1, r_2, \cdots, r_M)$. Through the logits of $\mathcal{Z}$, Zooter can be interpreted as reward prediction based routers. ERP, in turn, can be seen as a simplified and more scalable version of Zooter that directly regresses reward for each individual model as opposed to a single collective prediction across models.

- **Same model.** Route every prompt to a fixed model.

- **Random permutation.** Randomly reorders the model routing predictions of our expected reward predictor. Removes the conditioning on prompt $x$ but preserves the predicted best model distribution of our expected reward predictor and hence the incurred cost as well.

- **Purely random.** Route prompts to random models.

- **Per-category best (oracle).** Route each prompt in a category to the model that had the best average cost-adjusted reward in that category in the training set. Requires oracle knowledge of the prompt categories at inference time.

Notably, we omit a preference-based router because a 5-class preference model would be significantly more complex to train than the ERP-based policy, and because performance was comparable to direct win prediction in our pairwise experiment. Extending the ERP-based policy to five models, on the other hand, uses the same components that we used in the pairwise setting, with no additional training.

**Results.** We evaluate each routing policy based on its ability to trade off prompt-averaged cost and regret effectively. In Figure 1(b) and Figure 6, we plot the Pareto frontier induced by different exchange-rate values $\lambda$, for the OpenAssistant-RM, GRM-2B-RM, and InternLM-RM rewards, respectively. For all reward models, the ERP and Zooter routers are able to establish a Pareto frontier that contains all non-oracle baselines. The comparison to the per-category best oracle (black diamonds), which has access to oracle prompt category labels at test time, is also compelling. Even without category labels, ERP and Zooter are able to provide similar cost-performance tradeoffs across all three reward models. For OpenAssistant-RM and InternLM-RM, ERP dominates the oracle, while for GRM-2B-RM, the oracle breaks the Pareto front at one point, earning slightly lower regret at a higher cost at one point. In part, this reflects the fact that for this reward model, within some categories, most wins went to a single model, so routing all queries in those categories to a single model is nearly optimal.

Given the similarity in effectiveness and increased simplicity of ERP in relation to Zooter, we may reasonably infer the success of Zooter is inherently from the predictability of expected reward. In addition, the scalability of ERP with respect to the number of models is more favorable since only responses and rewards from a new model are needed to add it to an ERP router.

## 5 Discussion and Limitations

The main result of this paper is to demonstrate a useful property of LLMs and reward models: the per-prompt expected reward for a given model is readily predictable, and can serve to lift reward functions on responses to be reward functions on models. When expected rewards are predictable, downstream tasks that might otherwise require preference-level comparison data can be achieved effectively by predicting model-level reward distributions in isolation. While we explored model routing as the primary application in this paper, many other inference-time applications could be possible, including, for example, hot-swapping system prompts.

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

# A APPENDIX

## A.1 ERP FOR VERIFIABLE REWARDS

To test the generality of our approach beyond preference-based rewards, we evaluated ERP on tasks with objective correctness using the MMLU-Pro dataset (Wang et al., 2024b) (12000 examples across multiple disciplines). Our model pool consisted of Llama 3.1 8B, Llama 3.1 70B, Qwen 2.5 7B, and Qwen3 235B A22B. We used a binary reward function: 1 for correct multiple-choice answers, and 0 otherwise. The results in Figure 5 show that ERP remains effective in this setting, where we establish a strong Pareto frontier that outperforms baselines including Zooter. Notably, ERP nearly matches the per-category oracle baseline without requiring knowledge of prompt categories at inference time. These findings demonstrate that expected reward predictability extends beyond potentially biased reward models.

## A.2 ADDITIONAL FIGURES

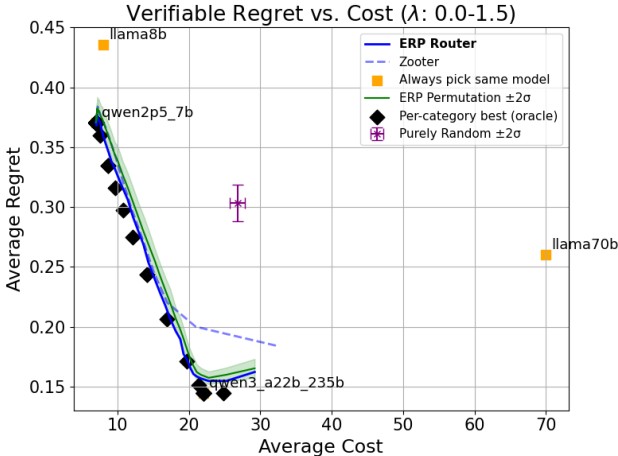

Figure 5: Regret-cost tradeoff for verifiable rewards on MMLU-Pro. ERP (blue) achieves a near-optimal Pareto frontier, closely matching the per-category oracle (black diamonds) and outperforming other baselines.

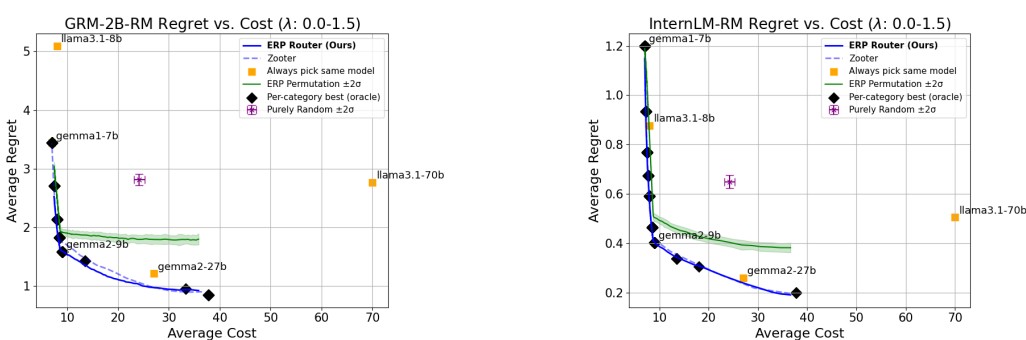

Figure 6: Regret-cost tradeoff curves using GRM-2B-RM and InternLM-RM on open-perfectblend.

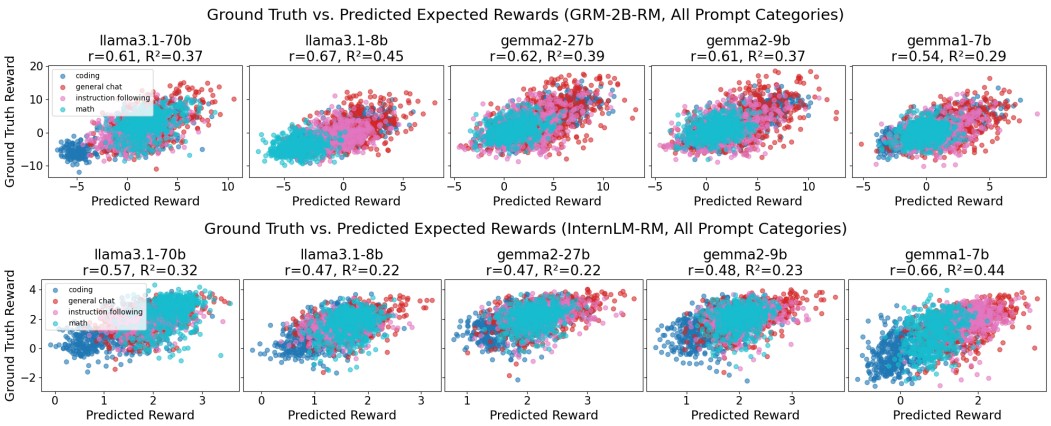

Figure 7: Predicted vs. actual reward on our test split of the open-perfectblend dataset for the GRM-2B (top row) and InternLM (bottom row) reward models for our set of five generating models (columns).

Table 2: $R^2$ of expected reward prediction within each open-perfectblend category of prompts using GRM-2B-RM and InternLM-RM.

| Model Name | Aggregate | Coding | Math | Instruction Following | General Chat |
|---|---|---|---|---|---|
| **GRM-2B-RM** | | | | | |
| llama3.1-70b | 0.37 | 0.55 | 0.18 | 0.11 | 0.30 |
| llama3.1-8b | 0.45 | 0.60 | -0.21 | 0.09 | 0.24 |
| gemma2-27b | 0.39 | 0.61 | 0.01 | 0.19 | 0.29 |
| gemma2-9b | 0.37 | 0.59 | 0.03 | 0.16 | 0.28 |
| gemma1-7b | 0.29 | 0.51 | 0.01 | 0.12 | 0.27 |
| **InternLM-RM** | | | | | |
| llama3.1-70b | 0.32 | 0.36 | 0.12 | -0.14 | 0.15 |
| llama3.1-8b | 0.22 | 0.26 | 0.09 | -0.03 | 0.18 |
| gemma2-27b | 0.22 | 0.19 | 0.04 | 0.02 | 0.25 |
| gemma2-9b | 0.23 | 0.17 | 0.02 | 0.01 | 0.21 |
| gemma1-7b | 0.44 | 0.23 | 0.09 | 0.06 | 0.24 |

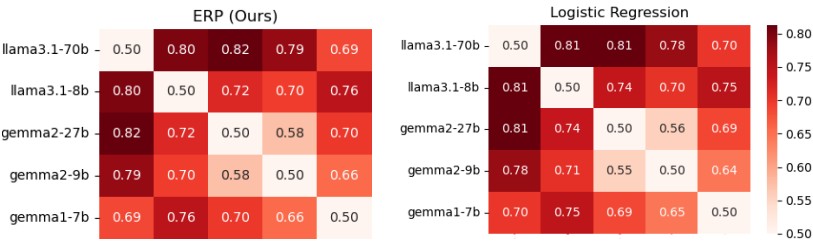

(a) AUROC scores using GRM-2B-RM.

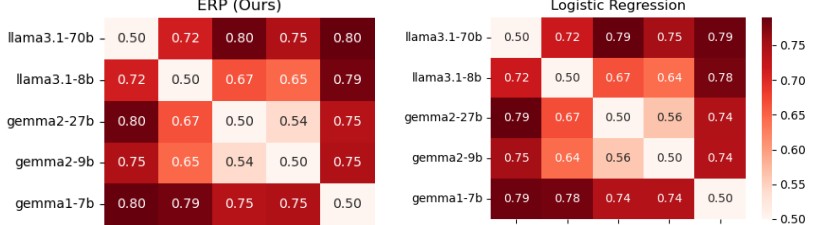

(b) AUROC scores using InternLM-RM.

Figure 8: Comparison of our expected reward predictor (ERP) to logistic regression when predicting pairwise model wins.

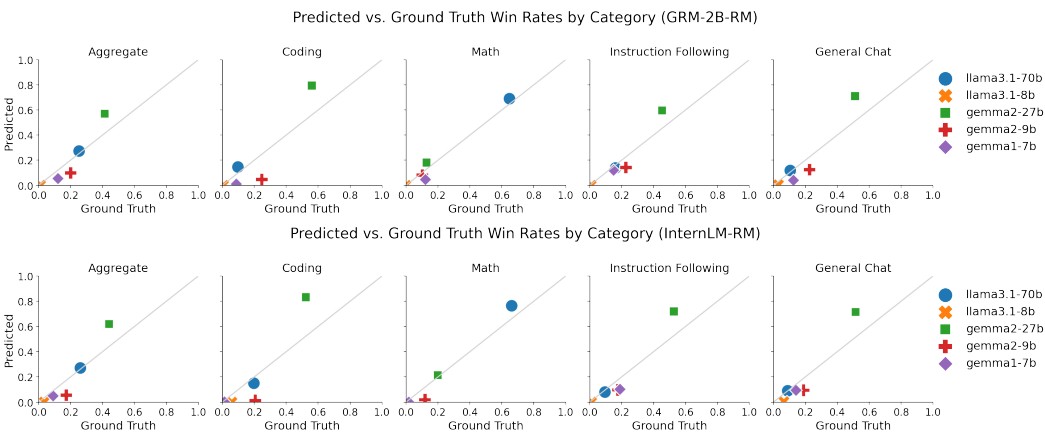

Figure 9: Ground truth and predicted win rates (among 5 LLMs) within each open-perfectblend category, according to GRM-2B-RM and InternLM-RM.

