# OpenReview forum: "Expected Reward Prediction, with Applications to Model Routing"
_ICLR.cc/2026/Conference — Submitted to ICLR 2026_

### Official Review · Reviewer_vmy7 · 2025-11-01

**Soundness:** 2
**Presentation:** 2
**Contribution:** 3
**Rating:** 4
**Confidence:** 4

**Summary:**

This paper studies whether you can predict the expected reward that a language model would get on a prompt before actually sampling from it. The authors predict the expected reward across different models via simple linear models ontop of prompt embeddings. They test this across 5 different generating models (Llama 3.1, Gemma 2, Gemma 1) and 3 reward models (OpenAssistant-RM, GRM-2B-RM, InternLM-RM) using the open-perfectblend dataset. As an application, they use these predictions for model routing, where each prompt is sent to the model predicted to give the highest expected reward. Their Expected Reward Prediction (ERP) routing approach outperforms dedicated routing methods such as Zooter.

**Strengths:**

* The idea of predicting statistics of the response distribution is both interesting and novel and seems like it could open up more downstream applications.
* The predictive performance of a linear model trained ontop of independent model embeddings was surprising. That such a method can be a pareto improvement over the zooter router is an encouraging sign.

**Weaknesses:**

* The only downstream application of expected reward prediction explored in this paper is model routing. While prompt modification is mentioned it is not explored as an application. It would strengthen the paper greatly to demonstrate that there are other usages of expected reward prediction.
* The space of expected reward predictors is left unexplored in the paper. While its very interesting that the embeddings of an independent model can be used to make an accurate predictor, there are many questions left unanswered. As examples, how does predictive performance scale with the size of the embedding model or can using embeddings from the models being tested or the RM scoring samples lead to better performance?

**Questions:**

* Did you consider predicting properties other than the expected reward? It seems as though there are many other statistics of the response distribution that would be useful to predict for downstream applications.

---

> ### Author Response · Authors · 2025-12-03
>
> We appreciate the reviewer’s comments on strength of ERP and novelty of the method.
> - With regards to further applicability, we note results for MMLU-Pro (detailed in Appendix A.1) which showed that our ERP method works well even when the reward signal is objective accuracy rather than a reward model score.
> - Regarding the effectiveness of different prompt embeddings, in earlier experiments, we found that using the same model’s embedding (last token embedding from the prompt) for predicting the expected reward yielded no improvements when compared to the gte embedding model.
> - Furthermore, in earlier experiments, we had also tried predicting the variance of reward but it turned out to be difficult to predict in any meaningful way. For example, with Gemma1-IT 7B we observed $R^2=0.22$ on the open-perfectblend prompts when attempting to predict the variance of reward and it was not helpful on downstream applications such as more efficient BoN sampling.

---

### Official Review · Reviewer_eT2c · 2025-11-01

**Soundness:** 3
**Presentation:** 3
**Contribution:** 3
**Rating:** 8
**Confidence:** 4

**Summary:**

This paper reveals an interesting property of reward models: the ability to, within reason, predict the reward given by a reward model to a given response to a prompt from a specific model. They use this property to develop a model routing function using the predicted reward, and show strong results when routing between models from different model families.

**Strengths:**

This paper finds an unexpected phenomenon, and has well-designed experiments to critically analyze this phenomenon. The fact that rewards are reasonably predictable for given models has interesting implications, both for model routing as shown in this paper, and other potential applications (as they mention), such as test-time modifications to system prompts/etc, and has potential implications for building simpler or more efficient classical RMs (e.g. distilling a RM into a smaller classifier, assuming it can reasonably predict the outcome).

**Weaknesses:**

While the experiments make sense to me, I'd also be interested to see how well this works for creating a reasonable ensemble of models, and how they perform on different benchmarks. For example, we could see what the Alpaca Eval score is when routing prompts to different models, or how this setup performs on benchmarks targeting math, or instruction following. I don't think those would be required for this paper, but they would be 1) very interesting, and 2) would show the further applicability of this method.

**Questions:**

Like I stated above, I'd be curious to see how this method improves performance on some more "standard" benchmarks vs an oracle setting, or models in isolation, if possible.

---

> ### Author Response · Authors · 2025-12-03
>
> We appreciate the reviewer’s comments regarding the experimental design and interest in the phenomenon.
> - We agree that evaluating on other benchmarks like AlpacaEval would be a valuable extension of this work. We note results for MMLU-Pro (detailed in Appendix A.1) which showed that our ERP method works well even when the reward signal is objective accuracy rather than a reward model score.

---

### Official Review · Reviewer_VF8L · 2025-11-01

**Soundness:** 3
**Presentation:** 3
**Contribution:** 3
**Rating:** 4
**Confidence:** 3

**Summary:**

The paper proposes a method to predict model suitability for given prompts, by trying to predict the expected reward from a reward model. It then applies their approach to a model routing setup, where different prompts are routed to different policy models according to their expected rewards. The results show that it is possible to predict expected rewards of models and that the routing approach outperforms other baselines.

**Strengths:**

- The paper is well written and generally clear.
- It’s great to see that a relatively small model is able to accurately predict expected rewards.
- The simplicity of the approach and some interesting findings such as that there is enough variability between models.

**Weaknesses:**

- Motivation: the paper doesn’t convincingly motivate why it is advantageous to predict the expected reward instead of the actual RM rewards.
- Practical applicability/downstream performance: I was missing experiments that showed how this routing setup would actually improve post-training of language models on downstream evaluations. All evaluations in the paper are intrinsic evaluations, but it would be interesting to see actual applicability to concrete tasks.
- Results: Some of the metrics in the tables are not appropriately explained and make it hard to judge how convincing the results actually are.

**Questions:**

- What is the benefit of being able to predict the expected reward from a prompt?
- Can the same approach be applied to llm-as-judge reward models?
- Since this is per prompt type: could you use the approach to predict how well models would do on certain benchmarks? I.e. maybe use it as an early intervention on post-training checkpoints?
- 215: Why do you need repeated samples at inference time when you just regularly use an RM?
- Could you explain the R2 metric in Table 1.
- 274 - 277: paper would be more substantial if you could provide such results
- Citation suggestion: Miranda, Lester James V., et al. "Hybrid Preferences: Learning to Route Instances for Human vs. AI Feedback."

---

> ### Author Response · Authors · 2025-12-03
>
> We thank the reviewer for their feedback and for highlighting the simplicity and effectiveness of our approach. We would like to clarify any confusion below:
> - The RM rewards depend on the model’s response - hence we take an expectation across model responses and in practice estimate the expectation via Monte Carlo sampling. As a result, ERP does in fact predict RM reward.
> - The primary goal of model routing is to select the best model before paying the computational cost of generating a response. If we used the RM at inference time, we would have to generate a response from every model in the pool first, score them all, and then pick the best one. Our method predicts the potential quality from the prompt alone, allowing us to route to a single model and save inference computation.
> - Repeated sampling is required only during the training phase to construct the ground-truth labels (the empirical mean reward). At inference time, we do not sample and simply run the prompt through the linear predictor to estimate the expected reward.
> - We agree that downstream tasks are important. While we do not focus on post-training in this paper, we point to Appendix A.1, where we evaluated our method on MMLU-Pro. In this setting, the reward is objective correctness (1 for correct, 0 for incorrect). Our ERP router successfully established a Pareto frontier, showing verifiable correctness prediction as a capability of our method.

---

### Official Review · Reviewer_XGcB · 2025-11-01

**Soundness:** 3
**Presentation:** 2
**Contribution:** 2
**Rating:** 4
**Confidence:** 3

**Summary:**

This paper investigates the predictability of the expected reward of an LLM given a prompt before the LLM generates any responses. Results show that the expected reward can be predicted via a simple linear model given a large amount of training data (predicted rewards by some reward model for a collection of LLM completions). This expected reward can then be used in model routing, selecting LLMs from a model pool for answering a new prompt. Improvements related to performance and cost are observed over existing routing baselines.

**Strengths:**

- The technical contents are clearly written, and the paper is generally easy to follow.
- The experiment scale is great, covering a good number of LLMs, reward models, and routing baselines.
- The empirical observations that the expected reward is readily predictable, and that the rewards (using the same reward model) are comparable across tasks and across LLMs, are quite valuable for the research community.
- The proposed method requires minimum computation *during test time*.

**Weaknesses:**

- Paper clarity: It would make sense to move more routing results with comparisons against the baselines to the main text of the paper. Currently there is only one figure with one reward model in Figure 1(b).
- Discussions on limitations and future works are quite thin.
- The non-trivial train-time computational costs of the proposed method for calculating expected reward are not adequately discussed. It seems to me that the predictability of the expected reward comes from the availability of a substantial amount of training data obtained from the reward model, and that the linear model is an oracle to the reward model. According to Section 3, one linear model is required per LLM per reward model. To train this, one would need access to a dedicated training dataset, sample the LLM for K (K=32 in the paper) generations per prompt in that dataset, use the reward model to get the ground-truth rewards, and then train the linear model. When one has 5 models in the LLM pool, this process needs to be repeated for each LLM. So, obtaining the linear model already seems very costly, given that the train/test split for this process is 50/50. I believe adding the train-time computational costs comparison between the proposed method and the compared routing baselines would make a fairer comparison and give readers a more comprehensive understanding.
- If we look at the scatter plots in Figure 2, the variability of predicted reward seems quite high - for a point with ground truth reward of 0, the linear model would give a predicted reward interval of something like [A, A+5] for many points. Could the authors please elaborate on this variability?

**Questions:**

Please see weaknesses above.

- What exactly are the advantages of the proposed method over the pre-category best baseline (with the help of a reward model at test time)?
- Given the procedure of training the linear model, how robust would the method be for predicting expected rewards for out-of-domain prompts? Would it always need to see similar prompts first in the training dataset?
- One additional comment: looking at regression-based reward models (for example those trained on helpsteer datasets) might give richer reward signals. Could be interesting future work.

---

> ### Author Response · Authors · 2025-12-03
>
> We appreciate the reviewer’s comments that the paper is well-written, easy to follow, and contains extensive experimental results. We clarify the reviewer’s concerns below:
> - While generating $K=32$ samples for training does incur a fixed upfront cost, this is significantly lower when compared to preference-based routers which typically require pairwise comparisons and scale quadratically with the number of models. Every new model would require preference data comparing it against the existing pool of models whereas with ERP the new data only needs to come from the model itself and scales linearly.
> - Regarding the scatter plots, while there is variance, the signal is strong enough to separate "good" models from "bad" models for a given prompt. As shown in our routing experiments, perfect precision is not required to outperform baselines. The predictor only needs to rank the models correctly relative to one another to achieve lower regret.
> - The per-category best baseline only uses the reward model on the train set to evaluate the best model per category according to cost-adjusted reward. At test time, this baseline simply routes each prompt to the best model according to the category of that prompt. ERP does not use the category of the prompt which may be unknown in practice at test time.

---

### Meta-Review · Area_Chair_91dT · 2026-01-08

**Summary:**

The original main concerns revolve around: what applications is this useful for beyond model routing, why not just use a reward model instead of trying to predict it and associated clarity issues (XGcB and VF8L did not seem to understand if multiple samples were required for this procedure), and high variability and noise in the shown scatter plots as well as related evaluation concerns particularly in OOD scenarios. On the positive side, the model routing issue is one that has been open in the field for a while with none managing to make significant progress so this line of work is engaging.

**Reviewer Concerns:**

Most of the concerns about the clarity of the Expected Reward Prediction procedure and whether sampling is needed (the second main category of issue from above) seem to have been addressed though not necessarily the others. In particular, issues surrounding generalization to OOD prompts (eT2c) and broader downstream evaluations are unaddressed. This has me leaning towards a reject overall given 3 borderline rejections and one strong acceptance review.

**Reviewer Scores:**

Multiple reviewers who gave borderline reject scores would be unlikely to increase given this rebuttal, in particular vmy7 and XGcB as they wanted more information on downstream applications etc. VF8L has a higher probability of increasing score to a 6 than the others as their questions on clarity were addressed at least.

eT2c had a high initial score but relatively little justification compared to the others and likely would not have moved their score either which way.

---

### Decision · Program_Chairs · 2026-01-26

Reject